# Fighting Cancer with Bacteria and Their Toxins

**DOI:** 10.3390/ijms222312980

**Published:** 2021-11-30

**Authors:** Dragan Trivanović, Krešimir Pavelić, Željka Peršurić

**Affiliations:** 1Department of Oncology and Haematology, General Hospital Pula, Santorijeva 24a, 52100 Pula, Croatia; dtrivanovic@obpula.hr; 2Faculty of Medicine, Juraj Dobrila University of Pula, Zagrebačka 30, 52100 Pula, Croatia; pavelic@unipu.hr; 3Faculty of Chemical Engineering and Technology, University of Zagreb, Trg Marka Marulića 19, 10000 Zagreb, Croatia

**Keywords:** bacteria, bacterial toxins, lung cancer, proteomics, extracellular vesicles

## Abstract

Cancer is one of the most important global health problems that continues to demand new treatment strategies. Many bacteria that cause persistent infections play a role in carcinogenesis. However, since bacteria are well studied in terms of molecular mechanisms, they have been proposed as an interesting solution to treat cancer. In this review, we present the use of bacteria, and particularly bacterial toxins, in cancer therapy, highlighting the advantages and limitations of bacterial toxins. Proteomics, as one of the omics disciplines, is essential for the study of bacterial toxins. Advances in proteomics have contributed to better characterization of bacterial toxins, but also to the development of anticancer drugs based on bacterial toxins. In addition, we highlight the current state of knowledge in the rapidly developing field of bacterial extracellular vesicles, with a focus on their recent application as immunotherapeutic agents.

## 1. Introduction

Cancer is a major disease burden and economic problem worldwide. With over 18 million cases in 2018, we can expect 29 million cases by 2040 due to aging and population growth around the globe. Here, we use lung cancer as a paradigm, although the effect of bacteria applies to other tumors as well. Lung cancer caused more deaths in 2017 than breast, prostate, colorectal, and brain cancer combined. It is estimated that lung cancer was the leading cause of cancer death in both genders in Europe in 2017, accounting for 24% of cancer deaths in males and 15% in females [1,2].

Tobacco smoking remains the leading cause of lung cancer. However, smoking addiction does not fully explain the higher lung cancer incidence rates recently reported in young women compared with men born around the 1960s. Several other factors have been detected as risk factors for lung cancer, including genetics, exposure to asbestos, radon, arsenic, and non-tobacco-related polycyclic aromatic hydrocarbons [3,4]. 

Lung cancer is generally divided into two histological pathological types: non-small cell lung cancer (NSCLC), which accounts for 80–90% of lung cancer cases, and small cell lung cancer (SCLC). Most patients with NSCLC have advanced, unresectable disease, as well as a high mortality rate, and they benefit little from standard therapy and have to limited treatment options. Unfortunately, there are still not many successful ways to treat patients with this type of tumor. Treatment options for lung cancer include surgery, chemotherapy, radiotherapy, and, more recently, targeted therapy and the emerging immunotherapy, which has significantly better outcomes. Therefore, there is an unmet need for a multidisciplinary approach to diagnostic and therapeutic procedures for all patients [3].

Although the cause of lung cancer is discussed in the context of smoking or exposure to environmental carcinogens, much attention has recently been paid to the role of the microbiome. The human body coexists with a complex microbiome that includes bacteria, fungi, viruses, and protozoa that colonize the host microenvironment and form a dynamic microecological system that has evolved over time [5]. A growing number of studies have profiled the microbiome in upper and lower respiratory samples from healthy adult lungs. The most abundant bacteria have been identified, including the phyla Bacteroidetes, Firmicutes, and Proteobacteria, and genera such as *Streptococcus*, *Pseudomonas*, *Veillonella,* and *Prevotella* [6,7,8]. About 10 species have been identified by the International Agency for Cancer Research (IACR) as agents that are carcinogenic to humans, and more than 16% of all cancers in the world can be attributed to infection with certain viruses, bacteria, and parasites [9,10]. The cancers induced by bacterial infections are listed in Table 1.

There has been increasing interest in the possible connection between the lung microbiome and lung cancer risk (Table 2). Although the underlying mechanisms still need to be clarified, studies have already shown that there is a strong link between microbiota dysbiosis and lung carcinogenesis [6,7,8,34,35,36,37,38,39,40]. 

In prostate, lung, colorectal, and ovarian cancer screening trials involving over 77,000 subjects, antibody titers for *Chlamydia pneumoniae* were significantly higher in patients with lung cancer in comparison to healthy subjects [41]. Furthermore, the use of antibiotics has also been linked with a risk of developing lung cancer [42]. In addition, studies have shown an association between lung cancer and *Mycobacterium tuberculosis* [11]. Overall, these studies suggest that microorganisms can contribute to lung carcinogenesis primarily by inducing inflammation. 

Despite the fact that bacteria can contribute to cancer development, on the other hand, bacterial cancer therapy has been recognized as one of the novel approaches in cancer treatment. Therefore, in this review, we focus on different strategies in bacteria-based cancer therapy, analyzing in particular their advantages and possible future research directions. Special attention is also paid to how proteomics, as a rapid and powerful omics discipline, can contribute to this field and how it could help address new challenges in the development of therapeutics based on bacterial toxins.

**Table 2 ijms-22-12980-t002:** Suggested connection between lung microbiome and lung carcinogenesis.

Bacteria (Family/Genus/Species)	Type of Sample	Correlation or Mechanism	Reference
*Granulicatella, Strepotoccocus, Abiotrophia*	Buccal-oral, sputum	Significant difference in presence and distribution	[43]
*Captocytophaga, Selenomonas,* *Veilonella*	Saliva	Significant difference in presence and distribution	[44]
*Veilonella,* *Megasphaera Actinomyces,* *Arthrobacter, Capnocytophaga* *Rothia,* *Streptococcus*	Bronchoalveolar lavage	Significant difference in presence and distribution	[45]
*Thermos,* *Ralstonia,* *Legionella*	Lung cancer vs health lung tissue	Significant difference in presence and distribution	[46]
*Streptococcus viridans, Granulicatella adiacens*	Sputum	Significant difference in presence and distribution	[46]
*Streptococcus intermedius, Mycobacterium tuberculosis*	Lung tissue and bronchoscopy samples	Significant difference in presence and distribution	[6]
*Streptococcus*	Bronchial brushing	Significant difference in presence and distribution	[37]
*Streptococcus,* *Veilonella*	Lung cancer vs health lung tissue	Significant difference in presence and distribution	[39]
*Streptococcus,* *Neisseria*	Lung cancer vs health lung tissue	Significant difference in presence and distribution	[47]
*Proctobacteria* *Firmicutes* *Bacteroides*	Lung tissue	Significant difference in presence and distribution	[48]
*Acidovorax,* *Klebsiella,* *Rhodoferax,* *Anaerococcus Cyanobacteria*	Lung tissue	Significant difference in presence and distribution	[39,49]
*Streptococcus,* *Prevotella*	Lung tissue	Significant difference in presence and distribution	[50]
*Sphingomonas, Blastomonas*	Saliva	Significant difference in presence and distribution	[51]
*Bradyrhizobium japanicum*	Bronchial brushing	Significant difference in presence and distribution	[52]
*Veillonella,* *Prevotella,* *Streptococcus*	Lung cancer tissue	Upregulation of ERK and PI3K signaling pathways	[39]
*Acidovorax,* *Klebsiella,* *Rhodoferax,* *Anaerococcus*	Lung cancer vs health lung tissue	Significantly higher abundance of in SCC than in adenocarcinoma	[36]
*Acidovorax*	Lung cancer vs health lung tissue	Abundant in patients with TP53 mutation-positive SCCLC and smoking history	[36]
*Bradyrhizobium japonicum,* *Acidovorax*	Bronchial brushing	Significant difference in presence and distribution	[52]
Bacteroidaceae, Lachnospiraceae, Ruminococcaceae	Lung cancer vs health lung tissue	Significant difference in presence and distribution	[38]
*Mycobacterium tuberculosis *	Lung cancer vs health lung tissue	Chronic inflammation	[11]

## 2. History of Using Bacteria in Cancer Therapy

The German physicians Busch and Fehleisen separately observed the regression of tumors in patients who had a skin infection caused by *Streptococcus pyogenes* [53]. Independently, in 1893, the surgeon William Coley discovered that a patient with sarcoma had fully recovered after an accidental erysipelas infection. He conducted clinical trials on patients with terminal cancers and recorded tumor regression after infection with killed bacterial species *Streptococcus pyogenes* and *Serrati marcescens*. Later, Coley developed “Coley’s toxin”, a vaccine made from these two bacterial species, which was widely used in various cancers to simulate infection by inducing fever, inflammation, and chills [54]. Coley’s toxins were not easy to produce or administer and were associated with side effects such as fever and negative outcomes, so they did not become the standard of care in cancer treatment. However, the early success of Coley’s toxins paved the way for today’s advances in bacteria-based cancer therapy. In 1976, bacterial cancer therapy with Bacillus Calmette–Guerin (BCG) was established by Morales, Eidinger, and Bruce, who successfully used attenuated *Mycobacterium bovis* to treat bladder cancer [55]. There have been a few other attempts to treat cancer with live bacteria (*Streptococci* and *Clostridia*) and with genetically engineered bacterial toxins in combination with other treatments (*Salmonella, Clostridium, Lactobacilli, E. coli, Bifidobacterium, Pseudomonas, Streptococcus, Proteus, Caulobacter,* and *Listeria*) [54].

## 3. Molecular Mechanisms of Anticancer Bacterial Action

There are different mechanisms of fighting cancer by bacteria (Figure 1). Stimulating inflammation as a consequence of the immune response promotes bacterial transmission to neoplastic tissue, which in turn promotes the production of inflammatory cytokines and subsequently leads to the inhibition of tumor growth [56]. The pathogenic interaction of bacteria enhances the immune system of the host in different ways and significantly increases the amount of inflammatory cytokines in tumors, which results in drastic tumor growth suppression. IL-1β is the proinflammatory cytokine that plays a pivotal role in immunity against pathogens [31]. Patients with high-diversity gut microbiomes display enhanced memory T cell and natural killer cell signatures in the periphery blood. Interestingly, while the intestinal microbiota can promote local inflammation and carcinogenesis of the gastrointestinal tract [57,58] on transplanted tumors at distal sites, they can exert an opposing effect by priming the host immune system and boosting the systemic antitumor immune response [59]. Recent data have shown that commensal microbiota can alter the outcomes of immunotherapeutic therapy in human cancers [60]. Moreover, fecal microbiota transplantation (FMT) from cancer patients responding to immune checkpoint inhibitors to sterile or antibiotic-treated mice improved the antitumor efficacy of the programmed cell death protein 1 blockade in mice [61]. In an experimental mouse model, Le Noci et al. showed that the aerosolization of bacteria isolated from lung microbiota of antibiotic-treated mice reduced lung metastasis implantation by improving the immune response against cancer. These changes are associated with the reversion of immunosuppression observed in the tumor microenvironment, favoring the immune response against cancer cells [62]. 

Another mechanism is through the depletion of nutrients required for cancer cell metabolism [31]. Systemic administration of *Salmonella* bacteria, which invade the solid tumor through a severe hemorrhaging area, leads to necrotic regions where the bacteria proliferate, colonize the tumor, and decrease the tumor proliferation. This causes the tumor cells in the center of the tumor to die due to the deprivation of nutrients and oxygen. 

Some substances secreted by bacteria, such as bacteriocins, have shown anticancer activity and may act as synergistic agents to anticancer drugs. Cancer cell membranes are predominantly negatively charged, so bacteriocins bind preferentially to cancer cell membranes than to normal cell membranes, which are not charged and are therefore selective for bacterial binding [31,63]. 

Bacteria can act as anticancer agents through biofilms, a primitive form of multicellular life that are common to opportunistic bacterial pathogens such as *Salmonella tyhimurium* [31]. Studies have revealed the different potential of biofilms in cancer treatment. For example, the formation of bacterial biofilm on cancer cells during the SOS response can lead to the disruption of metastasis. In addition, biofilm can influence the development of colon cancer by altering the cancer metabolome to produce a regulator of cellular proliferation [64]. 

Furthermore, bacteria can be used as carriers for cancer therapeutic agents in cancer therapy. Non-pathogenic *Bifidobacterium adolescentis* was used as a vector for the expression of endostatin within tumors and inhibited angiogenesis and local tumor growth [65]. Similarly, *Bifidobacterium longum* was used as safe and stable delivery system for endostatin in cancer gene therapy [66]. Another promising drug delivery system in cancer therapy is bacterial minicells, which are anucleate biomaterials produced by abnormal cell division [67,68]. Minicells have several advantages over conventional nanoparticles in cancer therapy, such as safety, biocompatibility, and high drug loading capacity. Therefore, minicells coated with antibodies were successfully used to encapsulate a wide range of chemotherapeutics and are currently being investigated in clinical trials for cancer therapy [67]. For example, doxorubicin-loaded minicells targeting epidermal growth factor receptors (EGFR) via Vectibix have been evaluated in a phase I clinical trial in adults with recurrent glioblastoma [69]. 

Advances in molecular techniques have opened up new possibilities in cancer therapy, including the development of genetically modified non-pathogenic bacteria. Genetically engineered bacteria are designed to express reporter genes, tumor-specific antigens, and cytotoxic proteins or anticancer agents. For example, *Salmonella typhimurium* serovar VNP20009 and *Clostridium butyricum* M55, which selectively colonize tumors, have been used as delivery vectors in mouse tumor models without causing severe immune responses or toxic side effects. In addition, the most promising results were obtained with *Clostridia* strains (*C. acetobutylicum* and *C. beijerinckii*) that were successfully engineered to express genes encoding the specific bacterial enzymes cytosine deaminase and nitroreductase or murine tumor necrosis factor alpha [70].

Not only bacteria can be used in cancer therapy—bacterial toxins and spores have also shown promising anticancer activity and represent a new strategy for cancer treatment. Therapeutic trials using *Clostridium* spores by the intravenous route rely on the oncolytic activity of the bacteria. The use of the spores of the anaerobic bacteria has an advantage in the bacterial therapy of cancer, as these spores can survive the hypoxia and poorly vasculated regions of the tumor, as well as the necrosis conditions of the tumor which are refractory to traditional therapy. Furthermore, *C. novyi* spores have been investigated in combination with radiotherapy, radioimmunotherapy, and other chemotherapy in experimental tumor models [54]. Bacterial toxins are another effective means of inhibiting the growth of cancer cells (Figure 1) [71]. Bacterial toxins with antitumor activity are presented in detail in the next section.

## 4. Bacterial Toxins for Cancer Therapy

### 4.1. Introduction to Bacterial Toxins

Bacterial toxins with antitumor activity have been recognized as alternative anticancer agents for the treatment of advanced solid tumors. Cancer cells often have a high number of tumor-specific antigens on the cell surface and bacterial toxins bind to these antigens on the cell surface and are subsequently activated. Bacterial toxins that have been used as cell-targeted toxins include diphtheria toxin (DT), *Clostridium perfringens* enterotoxin (CPE), and *Pseudomona*s exotoxin A (PE) [54]. Immunotoxins are fusions of tumor-specific antibodies or its fragments to bacterial toxins [72,73]. Table 3 lists immunotoxins approved or in phase II and III clinical trials. Tumor-selective ligands bind to a receptor on the target cell and when the complex is internalized, the toxin causes cell death. LMB2 is an example of an immunotoxin. It consists of the Fv fragment of an antibody fused to a truncated *Pseudomonas* exotoxin and it has shown clinical activity in hairy cell leukemia and T cells neoplasms. On the other hand, *Clostridium perfringens* enterotoxin binds directly to CLDN3 and CLDN4 receptors, which are upregulated in tumor cells, and it can significantly inhibit tumor development [54].

Diphtheria toxin (DT) from *Corynebacterium diphtheria* was selected for the generation of the first immunotoxin, called denileukin diftitox (ONTAK), because of this high toxicity [74]. DT is a 535 amino acid exotoxin that binds to the heparin-binding epidermal growth factor precursor (HB-EGF) on the cell surface. DT can be cleaved into two main fragments: the DTA and DTB. The fragment DTB intercedes entry into the cell by binding to surface receptors and subsequent translocation into the cytoplasm by undergoing endocytosis. The fragment DTA is in charge of cytotoxic enzymatic activity, causing the disruption of protein synthesis and cell death [75]. 

To attenuate the lethal effect of DT, a modified DT was developed in which the cell receptor-binding domain of the toxin was removed. The “receptor-less” recombinant DT385 is highly cytotoxic to several cancer cell lines due to the inhibition of protein synthesis and induction of apoptosis. DT385 was effective in reducing angiogenesis and regressing tumor masses, as well as inhibiting subcutaneous growth of Lewis lung carcinomas [54]. 

A protein consisting of the fragment A in combination with recombinant human IL-2 was successfully used to treat cutaneous T cell lymphoma, resulting in denileukin diftitox DAB389 becoming the first immunotoxin approved by the Food and Drug Administration (FDA) for the treatment of cutaneous T cell lymphoma. It was specifically redirected towards cancer cells in order to target the IL-2 receptor, which is highly expressed on malignant T cells. However, the use of the targeted toxin resulted in severe toxic effects such as blurred vision or impaired color vision, nausea, diarrhea, skin, muscle pain, flu-like symptoms, and, most notably, vascular leak syndrome. In 2006, the FDA added a black box warning label to denileukin diftitox [74]. 

Hasenpusch et al. investigated the therapeutic potential of BC-819, a plasmid DNA which encodes for the A-fragment of diphtheria toxin, for the treatment of lung cancer in mouse tumor models and found that aerosolized BC-819 complexed to branched polyethylenimine is capable of reducing growth only in tumors arising from the luminal part of the airways [76].

*Clostridium perfringens* enterotoxin, a pore-forming bacterial toxin, has been effectively used in cancer therapy, particularly in the treatment of colorectal cancer, and numerous studies have shown that it has certain anticancer effects [77]. Another toxin that has been studied on different murine and human cancer cell lines is *Pseudomonas aeruginosa* exotoxin A (PE), which has the ability to block protein synthesis in mammalian cells [74]. For the treatment of HER2-overexpressing tumors, the therapeutic agent HER2-Affitoxin consisting of modified *Pseudomonas aeruginosa* exotoxin A (PE 38) and HER2-specific Affibody has been used [78]. Hashimi et al. engineered a chimeric EGF protein fused to the truncated N-terminal domain fragment of exotoxin A [79]. This chimeric EGF-ETA toxin targets and inhibits EGFR-positive cancer cells, potentially allowing it to be used to target EGFR-positive tumors that are resistant to monoclonal antibodies. Truncated *Pseudomonas* exotoxin can have even wider use rather than just in the development of immunotoxins; interesting results were also achieved with a newly developed nanotoxin comprised of PE38-loaded silver nanoparticles [80]. The nanotoxin induced the apoptosis of breast cancer cells, thus showing promising prospects for biologically synthesized silver nanoparticles to be used as a delivery system for targeting toxins to cancer cells. Furthermore, streptolysin O from *Streptococcus* bacteria has been tested as a new class of suicide gene therapeutic [81].

**Table 3 ijms-22-12980-t003:** Ligand-based and antibody-based immunotoxins approved or in phase II and III clinical trials [73].

Immunotoxin.	Antigen	Target	Toxin	Cancer type	Pipeline	Clinical Trial Identifier or Reference
Diphtheria Toxin Based						
Denileukin Diftitox (DAB389IL2)	IL-2R	IL2	DT (DAB389)	Melanoma, hematological	2008. approved by FDA for CTCL	[82,83]
DAB8486IL2	IL-2R	IL2	DT (DAB486)	Hematological	Phase I/II	[84,85]
Tagraxofusp SL-401 (DT388-IL3)	IL-3R	Variant IL-3	DT	Hematological	2018 approved by FDA for BPDCN	[86]
Tf-CRM107 (transMID)	TfR	Transferrin	DT (CRM107)	CNS	Phase I, III	NCT 00088400 NCT 00083447 NCT 00052624
DAB389EGF	EGFR	EGF	DT (DAB389)	EGFR positive cancers	Phase I/II	[87]
UCHT1	CD3	Murine anti-CD3-bis Fv	DT	Hematological	Phase I/II	NCT 00611208 NCT 01888081
DT 2219ARL bispecific	CD19 and CD22	Anti-CD22, Anti-CD 19 (sFv)	DT (DAB389)	Hematological	Phase I/II	NCT 00889408 NCT 02370160
***Pseudomonas* Toxin Based**						
TP-38	EGFR	TGFalpha	PE	CNS tumors	Phase II	NCT 00104091 NCT 00074334
Moxetumomab pasudodotox	CD22	Murine anti-CD22 d sFv fragment 2	PE	Hematological	Phase I, II, III	NCT 02338050 NCT 01829711 NCT 02227108 NCT 01030536
LMB-2	CD25	Anti-CD25scFv fragment	PE	Hematological, skin cancers	Phase II	NCT 00077922 NCT 00080536 NCT 00295958 NCT 00924170 NCT 00321555 NCT 00002765
SS1P	Mesothelin	Murine antimesothelin dsFv fragmentr	PE (PE38)	Mesothelioma, Cervical, Head and Neck, Lung, ovarian cancers	Phase I/II	NCT 01362790 NCT 00024687 NCT 00006981
Oportuzumabmonotox (VB4-8454)	EpCAM	Humanized anti-EpCAM scFv fragment	PE	Head and Neck, Squamous cell, bladder cancers	Phase II/III	NCT 00462488 NCT 00272181 NCT 02449239

Abbreviations: IL-R, interleukin (receptor); DT, diphtheria toxin; CTCL, cutaneous T cell lymphoma; BPDCN, blastic plasmacytoid dendritic neoplasm; CNS, central nervous system; TfR, transferrin receptor; EGFR, epidermal growth factor receptor; TGF, transforming growth factor; PE, *Pseudomonas* exotoxin A; EpCAM, epithelial cell adhesion molecule; NCT, ClinicalTrials.gov Identifier.

### 4.2. Proteomic Analysis of Bacterial Toxins Targeting Cancer Cells 

Potential candidates for anticancer therapy should be thoroughly investigated and well characterized before therapeutic application. Bacterial toxins with promising anticancer activity include botulinum neurotoxin type A, diphtheria toxin, exotoxin A, and listeriolysin O [88]. Although most of these toxins have been extensively studied for many years, advances in proteomics have provided new insights into their structure and function [89]. 

Botulinum neurotoxin type A is one of the seven toxinotypes produced by *Clostridium botulinum,* which can be further differentiated into toxin variants. To date, eight subtypes of botulinum neurotoxin type A have been described according to sequence diversity and immunological properties [90,91]. The relevance of botulinum neurotoxin type A subtypes is currently not well understood, but the potential impact on therapeutic properties cannot be ignored. Subtypes are usually identified using conventional molecular biology techniques [92]. However, these methods are associated with analytical limitations and therefore cannot overcome some new challenges to improve our understanding of bacterial toxins. In many areas of biotoxin research, conventional methods are being replaced by more sensitive mass spectrometry (MS), which is capable of analyzing toxins in a high-throughput manner [93]. Mass spectrometry-based proteomics is a powerful tool in various aspects of bacterial research. Mass spectrometry systems are now widely used for bacterial identification, as they allow rapid and precise detection of bacteria. In addition, the MS methods are playing an increasing role in the study of toxins produced by the bacterium [94]. Morineaux et al. have developed a liquid chromatography–tandem mass spectrometry (LC–MS/MS) method coupled to an immunocapture step with antibodies to characterize subtypes of botulinum neurotoxin type A [95]. Identification of L chain peptides specific for botulinum neurotoxin type A was performed using a triple quadrupole mass spectrometer (QqQ) in multiple reaction monitoring (MRM) mode. The developed MS method allowed unambiguous identification of subtypes A1 to A8 in a rapid and efficient manner. Matrix-assisted laser desorption ionization-time-of-flight (MALDI-TOF) mass spectrometry is another MS technique used to differentiate botulinum toxinotypes. In addition to identifying botulinum toxinotypes by amino acid sequence, MALDI-TOF can also be used to determine the enzymatic activity of the toxin [94]. Indeed, MALDI-TOF detection of toxin-induced cleavage of strategically designed peptide substrates enabled the determination of the enzymatic activity of botulinum neurotoxins.

Proteomics is also critical for understanding how toxins affect the host cell proteome. Analysis of the proteome changes induced by the toxin listeriolysin O (LLO) showed that LLO remodels the host cell proteome primarily by down-regulation of protein levels [96]. Moreover, this toxin-induced proteome remodeling was associated with major changes in the host ubiquitylome. To identify the LLO-induced changes in the proteome, stable isotope labeling by amino acids in cell culture was performed prior to shotgun proteomics. In addition, a second experiment was performed using label-free quantitative shotgun proteomics to compare protein abundance in LLO-treated and untreated cells. 

Proteomics has contributed to the unambiguous differentiation of toxinotypes and their subtypes and to a better understanding of their mode of action. However, the more important role of proteomics in cancer research is in defining the anticancer molecular mechanisms of bacterial toxins and the factors that influence the biological activity of the toxin or therapeutic proteins derived from toxins (Table 4). One of the most important contributions of next-generation proteomics is the detection of post-translational modifications of proteins (PTMs) that affect both protein homeostasis and the cellular processes in which they participate. Bacteria contain many different types of PTMs, such as oxidation, acetylation, phosphorylation, and S-thiolation that can potentially affect toxins and modulate their function [89]. Furthermore, post-translational modifications can occur during the production and storage of most therapeutic proteins derived from bacterial toxins. Moxetumomab pasudotox is an immunotoxin Fv fusion therapeutic protein derived from a 38-kDa truncated *Pseudomonas* exotoxin A (PE38) designed for the treatment of B-cell malignancies. In the production of moxetumomab pasudotox, a common posttranslational modification, asparagine (Asn) deamidation, was noticed. Since Asn deamidation could potentially affect the binding interface and biological activities of protein therapeutics, it is important to monitor these changes in the current producing process of clinical and commercial supplies. Lu et al. have shown that deamidation affects the biological activity of moxetumomab pasudotox by causing the conformational changes in the catalytic domain of the *Pseudomonas* exotoxin A region [97]. The effect of deamidation was monitored by both differential scanning calorimetry and hydrogen/deuterium exchange mass spectrometry. Hydrogen/deuterium exchange MS allowed the localization of the conformational change caused by deamination and thus confirmed itself as one of the most powerful techniques for monitoring the conformation and dynamics of therapeutic proteins. MS is also an essential tool in the discovery and development of anticancer drugs. Bachran et al. investigated modifications of an anticancer fusion protein consisting of anthrax lethal factor and the catalytic domain of *Pseudomonas* exotoxin A [98]. Mutation of the N-terminal amino acids and reductive methylation were performed to improve the efficacy of this therapeutic fusion protein on target cells. The success of a reductive methylation reaction to dimethylate all lysines was confirmed by electrospray ionization MS. The two tested modifications increased cytotoxic activity and improved its stability by preventing ubiquitination and subsequent proteasomal degradation. Furthermore, in another study, tandem MS methods were used to investigate how substitutions of individual residues within the exotoxin A domain III alter the structure, processing, and immunogenicity of this domain [99].

The search for new potential candidates with anticancer activity is a challenging process due to the complexity of the disease. Advances in proteomics contribute to many areas of cancer research, and, together with other omics disciplines, provide new insight into the biology underlying disease processes [101]. Proteomics deals with the identification and quantitative analysis of differentially expressed proteins under healthy and disease conditions. It plays an important role in identifying therapeutic targets, but is also useful for guiding treatments of molecular targets or evaluating therapeutic responses [102]. Zhao et al. used high throughput technology to investigate the antitumor activity of LukS-PV, a new potential bacterial toxin candidate for the treatment of human hepatocellular carcinoma [100]. LukS-PV is one of the two subunits of Panton–Valentine leukocidin (PVL), a toxin produced by *Staphylococcus aureus.* Ultra-performance liquid chromatography coupled with the quadrupole time-of-flight (UPLC-QTOF) technique was used to compare the protein expression profiles of LukS-PV-treated human hepatocellular carcinoma cell lines (HepG2) and control cells. The authors concluded that LukS-PV exerts an antitumor effect in hepatocarcinoma cells and identified a panel of dysregulated proteins associated with central carbon metabolism in cancer. 

## 5. Bacterial Extracellular Vesicles as Model to Fight Cancer

Bacterial membrane vesicles, which are enveloped in a lipid bilayer and carry toxins, virulence factors, nucleic acids, and metabolites, among other substances, are the subject of intense scientific investigation and represent a new approach for effective cancer treatment [103,104]. 

Bacterial extracellular vesicles (BEVs), also known as outer membrane vesicles (OMVs), are spherical, nanometer-sized, membrane-enveloped particles released by bacteria into the extracellular environment [105,106]. The name OMVs comes from the fact that they were first discovered in Gram-negative bacteria, although further scientific research has shown that BEVs are released by all classes of microbes, including Gram-positive bacteria [103]. In this review, for consistency with the publications cited herein, the name OMVs is used for extracellular vesicles of Gram-negative bacteria. Deciphering all the functions of BEVs is an ongoing, challenging process, even though it is already clear that BEVs play multiple roles in both cooperative and competitive strategies, such as function in bacteria—bacteria and bacteria—host communications, antibiotic resistance, and biofilm formation and survival [107]. 

BEVs carry multiple parent bacteria-derived components including membrane-bound and periplasmic proteins, enzymes, toxins, bacteria-specific antigens, polysaccharides, nucleic acids (DNA and RNA), lipoproteins, peptidoglycan, and others [104,106]. However, the composition of their molecular cargo can vary considerably due to different biogenesis pathways, growth conditions, and the genetic background and structure of the membrane envelope of the parent bacterium [106,108]. Mass spectrometry-based analysis of proteins in Gram-negative bacterial OMVs vesicles revealed thousands of OMV-associated proteins and helped clarify the biogenesis and pathophysiological functions of OMVs [109]. These proteomic workflows are usually composed from three main parts: first, the isolation of OMVs is preformed, then the mass spectrometry techniques are used to analyze OMVs and, finally, the systematic approach is applied to identify proteins. For example, Hong et al. used the isobaric labeling method together with tandem mass spectrometry to study quantitative changes in the proteome of OMVs from uropathogenic and probiotic *Escherichia coli* strains [110]. The same approach was also applied to determine the effects of the purification method and growth conditions on the proteome of OMVs. Isobaric tags for relative and absolute quantitation (iTRAQ)-based proteomic analysis revealed proteins involved in pathogenicity and proteins that could be markers of purity and culture conditions. 

Due to their vesicular structure and molecular cargo composition, OMVs have been intensively considered for various therapeutic applications, including possible use in cancer treatment (Figure 2) [104]. Initially, OMVs attracted considerable interest as bacterial vaccines and as drug delivery vehicles for cancer therapy [104,111,112,113,114,115,116]. More recently, the potential use of OMVs as immunotherapeutic agents for the treatment of various cancers has been explored [105]. Kim et al. showed that OMVs induce long-term antitumor immune response in mice bearing a tumor, and that interferon-γ (IFN-γ) plays an important role in mediating this antitumor response. Additional experiments revealed that surface proteins of OMVs are instrumental in triggering IFN-γ production. To further enhance their immunotherapeutic efficacy, OMVs were genetically modified by inserting the ectodomain of programmed death 1 [117]. This genetic modification enabled the inhibition of immune checkpoints and thus extensive regulation of the tumor immunological microenvironment. Furthermore, the study by Kuerban et al. demonstrated that OMVs can even play a dual role in cancer therapy [114]. OMVs were successfully used to transport the antineoplastic drug doxorubicin into human non-small-cell lung carcinoma cell lines. In addition, OMVs triggered appropriate immune responses, thus enhancing the anti-tumor effect of doxorubicin without observed toxicity. Another interesting example of the multiple functions of OMVs in cancer treatment is presented in a study in which an improved targeted photothermal cancer therapy was developed [118]. A single injection of OMVs not only activated the immune system but also led to the extravasation of red blood cells into the tumor, resulting in increased intratumoral optical absorption that facilitated effective photothermal ablation of tumors by a near-infrared laser. Nevertheless, before further clinical use of BEVs, some critical aspects should still be clarified, such as the method for mass production that will also ensure their safety in clinical trials [104].

## 6. Conclusions and Future Perspective

The dual role of bacteria in cancer cannot be disputed. Although bacterial infection is in the background of many cancers, the success of bacteria in fighting cancer has made bacteria-based cancer therapy a unique therapeutic option worth considering. The initial limitations in using bacteria as therapeutics have been overcome with the development of sciences such as cancer biology, microbiology, and bioengineering that have spurred the development of advanced bacterial therapies. Live or attenuated bacteria with different vectors or genetically engineered bacteria have emerged as potential strategies for cancer management with lesser toxicity for normal cells and better efficacy for treatment. Furthermore, therapy with bacterial toxins has shown promise because of their efficiency and specificity toward cellular molecules and signaling pathways. Bacterial toxins are used to develop immunotoxins by fusion to specific antibodies that are directed against tumor cells. Proteomics, as one of the omics disciplines, plays an inevitable role in the advancement and monitoring of therapeutics based on bacterial toxins. Mass spectrometry techniques not only enable precise identification of protein structures, but are also irreplaceable in monitoring modifications that can occur in proteins, thereby affecting their function. In addition, with advances in instrumentation, throughput methods with high selectivity and accuracy have been developed, allowing the analysis of complex proteomic profiles in a short time.

Bacterial anticancer therapy shows many other advantages, such as high tumor selectivity and the ability of bacteria to move from vasculature to tissues. In addition, bacterial therapy in combination with the conventional chemotherapy and radiotherapy or recently targeted and immunotherapy has shown higher efficacy in cancer-directed treatment, as well as better quality of life for the patients [54]. Unfortunately, clinical trials have shown that the side effects of bacterial therapy cannot be ignored. The efficiency of bacterial toxins is often high and healthy cells may also be harmed as serious side effects. Further limitations of bacterial anticancer therapy are the immunogenicity of the bacteria together with induction of septic shock due to high immunogenicity. Additionally, direct intra-tumoral injection may be necessary to achieve complete response. Whether bacterial extracellular vesicles represent a novel strategy in bacteria-based therapy that will overcome current limitations remains to be discovered. BEVs have been successfully used as drug delivery vehicles for cancer therapy, and more recently as immunotherapeutic agents. However, all bacterial-based therapy strategies require further research to improve efficacy and to minimize adverse effects before they can be used in clinical practice. The implementation of different omics approaches in further investigation is essential, as they provide significant insight at the molecular level.

## Figures and Tables

**Figure 1 ijms-22-12980-f001:**
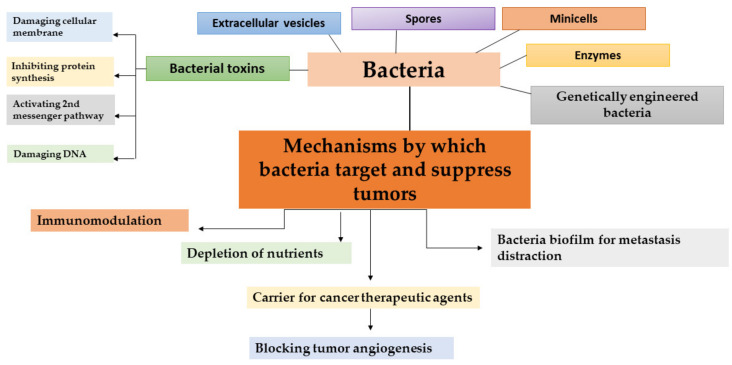
Different strategies in bacteria-based cancer therapy together with mechanisms by which bacteria and bacterial toxins target and suppress tumors.

**Figure 2 ijms-22-12980-f002:**
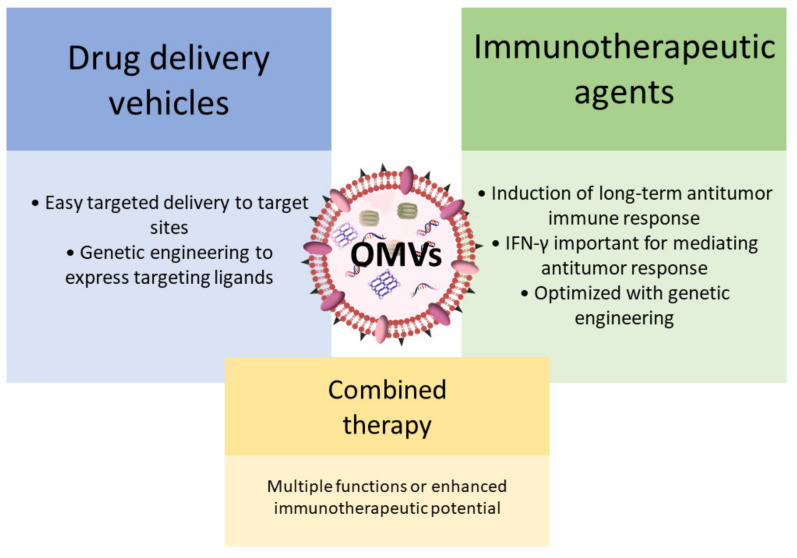
The use of OMVs in cancer therapy.

**Table 1 ijms-22-12980-t001:** Cancers induced by bacterial infection.

Bacteria	Cancer	Reference
*Mycobacterium tuberculosis*	Lung cancer	[11,12]
*Chlamidia pneumonia*	Lung cancer	[13]
*Salmonella enterica,* subsp *typhi*	Gallbladder cancer	[14]
*Prrphyromonas gingivalis*	Oral cancer Pancreatic cancer	[15,16,17]
*Fusobacterium nucleatum*	Oral cancer	[15,16,18]
*Treponema denticola*	Oral cancer	[19]
*Streptococcus anginosus*	Oral cancer	[19]
*Helicobacter pylori*	Gastric cancer, Mucosa-associated lymphoid tissue lymphoma	[20,21,22,23]
*Bacteroides* *fragilis*	Colon cancer	[24,25]
*Fusobacterium nucleatum*	Colon cancer	[26,27,28]
*E.coli*	Colon cancer	[29]
*Campylobacter jejuni*	Small intestinal lymphoma	[30]
*Citrobacter rodentium*	Colon cancer	[31]
*Chlamydia psittaci*	Cervical cancer, ocular lymphoma	[32]
*Citrobacter rodentium*	Colorectal cancer	[31]
*Streptococcus bovis*	Colorectal neoplasia	[33]

**Table 4 ijms-22-12980-t004:** The role of proteomics in the study of bacterial toxins with anticancer activity.

Sample Type	Toxin	Purpose of the Study	Mass Spectrometry (MS) Technique Applied in the Study	Reference
Crude culture supernatants, biological and food samples artificially spiked with culture supernatant of each *C. botulinum* A subtype	BoNT	Characterization of BoNT type A subtypes	HPLC-ESI-IT HPLC-QqQ	[95]
Cell cultures incubated with LLO	LLO	Identification of host proteome alterations induced by the LLO	HPLC-LTQ Orbitrap RSLCnano-Q Exactive high-field hybrid Quadrupole-Orbitrap	[96]
Moxetumomab pasudotox	PE (PE38)	Examination of structural and biological impact of deamidation	Hydrogen-deuterium exchange MS	[97]
Anthrax toxin fusion protein	PE	Examination of modifications of the therapeutic fusion protein that were predicted to improve its potency on target cells	ESI-MS	[98]
Exotoxin A domain III	PE	Identification how deimmunizing mutations alter structure, processing and immunogenicity of PE-III	NanoLC-Orbitrap Fusion MS	[99]
Cell lines HepG2 treated with LukS-PV	PVL	Identification of differentially expressed proteins to better understand antitumor activity of LukS-PV	NanoUPLC-QTOF	[100]

Abbreviations: BoNT, botulinum neurotoxins; HPLC, high-performance liquid chromatography; ESI-IT-MS, liquid chromatograph ion trap; QqQ, triple quadrupole; LLO, listeriolysin O; LTQ, linear trap quadropoly; RSLC, rapid separation liquid chromatography; PE38, 38-kDa truncated *Pseudomonas* exotoxin A; HepG2, human hepatocellular carcinoma; PVL, Panton–Valentine leucocidin; UPLC-QTOF, ultra-performance liquid chromatography coupled with quadrupole time-of-flight.

## Data Availability

Not applicable.

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
