# Peer review of "Fighting Cancer with Bacteria and Their Toxins"

_ijms, 2021, doi:10.3390/ijms222312980_

Round 1

Reviewer 1 Report

Similar review articles were published in the last three years:

PMID: 30950210

PMID: 31499068

PMID: 31827064

PMID: 32908523

And the latest one which published May 2021: PMID: 33303401

Why there is a need for another review paper?

I don't understand the Introduction section of the review.  It doesn't really match up with the title of the review.  The whole intro focuses on cancer and why cancer needs to be studied and it specifically focuses on lung cancer.  I don't know how relevant this is to the review article, especially just focusing on lung cancer.  You also discuss how bacteria are involved in cancer, specifically lung cancer.  I don't think all of this is needed.  I really like Section 2 as a nice intro.  That's really what you are talking about.  I thought that section was very germane to the title/topic of the paper.

I liked figure 1.

Line 167, I would delete the word "The" and just say "Introduction to bacterial toxins"

Line 169 and 171 should probably have a citation(s).

Table 3 spans 3 pages and the whole top row that defines what the columns are, are on a separate page.  Maybe the publisher could fix this, but it's annoying trying to read a table when you can't see what the columns represent.  On the pdf, it's fine, but on printed paper it's not good.  

In general, I really like how the tables were laid out, the references are easy to find.  

I thought that section 5 could use a diagrams/micrographs showing the pathway and/or bacterial vesicles.  

That last paragraph in the section, starting on line 368, that section could maybe use a diagram.  Are OMVs used primarily for drug delivery or for triggering/activating pathways with their surface proteins?  Maybe discuss the modification more or have a table showing modifications and cargo delivery composition.  I really like this paragraph, I think this paragraph is very relevant to the review paper.  I wish more focus/explanation/clarification was given to it. 

Line 363, Escherichia coli should be italicized

Line 338 and 339 you say that bacterial extracellular vesicles (BEVs) are also know as outer membrane vesicles (OMVs).  But then for the rest of the section you go back and forth between the two.  If they are the same thing, why not just choose one over the other and stick with that?  Also, along those lines, you define what the acronym BEV/OMV stands for, but then you continue to spell out the whole phrase.  Why not just use the acronym?

In regards to the Gram-negative bacteria that produce BEVs, are their certain ones that are predominantly used?  On line 390-391, you mention that one of the bottlenecks is mass producing BEVs or at least the method needs to be clarified (I'm not sure how to interpret that), but are there certain bacteria that are used more than others or can any Gram-negative be used?  Would the methods all be the same?  

Line 393, I like this sentence it's a nice call back to the Introduction section, but again I would shorten the Intro section and focus more on bacterial OMV and toxins, not lung cancer.  

That second sentence, "Although bacterial...", that would be good to have in the intro section because it really speaks to what the review paper is all about.

Author Response

  1. Similar review articles were published in the last three years: PMID: 30950210, PMID: 31499068, PMID: 31827064, PMID: 32908523. And the latest one which published May 2021: PMID: 33303401. Why there is a need for another review paper?

Reply:  Indeed, many reviews on bacteria or bacterial toxins have been published recently. However, the aim of this review is not only to provide an overview of the use of bacteria and bacterial toxins as anticancer agents in cancer therapy. The focus is on advantages and prospective directions in using different bacteria-based cancer therapies, especially bacterial toxins and BEVs. We also aimed to provide an overview of how proteomics, as a fast and powerful omics discipline, contributes to this field and how it could help address new challenges and improve our understanding of therapeutics based on bacterial toxins or BEVs. We have found that a broader perspective on this topic is lacking. This is now better explained in the Introduction section.

  1. I don't understand the Introduction section of the review.  It doesn't really match up with the title of the review.  The whole intro focuses on cancer and why cancer needs to be studied and it specifically focuses on lung cancer.  I don't know how relevant this is to the review article, especially just focusing on lung cancer.  You also discuss how bacteria are involved in cancer, specifically lung cancer.  I don't think all of this is needed.  I really like Section 2 as a nice intro.  That's really what you are talking about.  I thought that section was very germane to the title/topic of the paper.

Reply: The part of  introductory section about diagnosis and treatment of lung cancer has been shortened. However, we think that the introduction should give a broader picture of the role of bacteria in cancer. Since many cancers are induced by bacterial infections, we prefer to explain this complex topic by example. In doing so, lung cancer was chosen as a good example of how bacteria are involved in carcinogenesis.

  1. I liked figure 1.

Reply: Thank you very much for your remark.

  1. Line 167, I would delete the word "The" and just say "Introduction to bacterial toxins"

Reply: This was corrected

  1. Line 169 and 171 should probably have a citation(s).

Reply: The citation is [56] (line 174).

  1. Table 3 spans 3 pages and the whole top row that defines what the columns are, are on a separate page.  Maybe the publisher could fix this, but it's annoying trying to read a table when you can't see what the columns represent. On the pdf, it's fine, but on printed paper it's not good.

 Reply: Yes, indeed. The Table 3 was not positioned properly. We have deleted a part of the text and now the table is just on two pages. We are sure the publisher will fix this problem before publication.

  1. In general, I really like how the tables were laid out, the references are easy to find.  

Reply: Thank you for this comment.

  1. I thought that section 5 could use a diagrams/micrographs showing the pathway and/or bacterial vesicles.  That last paragraph in the section, starting on line 368, that section could maybe use a diagram.  Are OMVs used primarily for drug delivery or for triggering/activating pathways with their surface proteins?  Maybe discuss the modification more or have a table showing modifications and cargo delivery composition.  I really like this paragraph, I think this paragraph is very relevant to the review paper.  I wish more focus/explanation/clarification was given to it. 

Reply: We have added figure with diagram for section 5. We agree with the reviewer that the section on the use of OMVs in cancer therapy is very relevant to our review paper. OMVs have been studied both as vehicles for drug delivery and as immunotherapeutic agents that activate pathways. However, as this paper is already very long and covers several aspects, we would prefer not to make it longer.

  1. Line 363, Escherichia coli should be italicized

Reply: This was corrected

  1. Line 338 and 339 you say that bacterial extracellular vesicles (BEVs) are also know as outer membrane vesicles (OMVs).  But then for the rest of the section you go back and forth between the two.  If they are the same thing, why not just choose one over the other and stick with that?  Also, along those lines, you define what the acronym BEV/OMV stands for, but then you continue to spell out the whole phrase.  Why not just use the acronym?

Reply: Thank you very much for this comment. We have changed the entire wording to an acronym. In the scientific literature, there is no uniform name for extracellular vesicles of bacteria. We think that name "bacterial extracellular vesicles" (BEVs) implies extracellular vesicles of both Gram-positive and Gram-negative bacteria. However, in this review, for consistency with the cited publications, the name "outer membrane vesicles" was used for extracellular vesicles of Gram-negative bacteria.

  1. In regards to the Gram-negative bacteria that produce BEVs, are their certain ones that are predominantly used?  On line 390-391, you mention that one of the bottlenecks is mass producing BEVs or at least the method needs to be clarified (I'm not sure how to interpret that), but are there certain bacteria that are used more than others or can any Gram-negative be used?  Would the methods all be the same?  

Reply: Salmonella and E. coli have been mostly utilized to produce BEVs in scientific research. However, BEVs from various strains of bacteria showed efficacy in antitumor therapy. Some attempts to produce BEVs in larger quantities have been reported, but there is still a need to develop a method for mass production that will ensure safe clinical use.

  1. Line 393, I like this sentence it's a nice call back to the Introduction section, but again I would shorten the Intro section and focus more on bacterial OMV and toxins, not lung cancer.  

Reply: The Introduction was corrected. The paragraph on lung cancer has been shortened and the focus of the paper has been better explained.

  1. That second sentence, "Although bacterial...", that would be good to have in the intro section because it really speaks to what the review paper is all about

Reply: The similar sentence was added to introduction.

Reviewer: 2

  1. The authors of the paper titled “Fighting cancer with bacteria and their toxins” have presented a review featuring the functions of bacteria, their toxins and their released extracellular vesicles in tackling cancer from therapeutic as well as immunomodulatory perspectives. The review is interesting to a broader audience and is fit for publication with minor modifications.

Reply: We would like to thank you very much for your helpful remarks and suggestions. We tried to consider all of them and introduce suitable changes in the revised manuscript. We hope that now the new version of the manuscript will be recommended for publication. The detailed answers are below.

  1. The review has a lot of relevant tables but is lacking any pictorial depictions of the different mechanisms leading to cancer cell death. e.g- increase in intracellular ROS production by bacterial toxins, modulation of tumor microenvironment by different cytokines etc. This should be included.

Reply: Different mechanisms by which the bacterial toxins act are added to Figure 1. The different toxins have different modes of action to destroy the cancer cells, and therefore we have chosen to write them rather than show them pictorially, as Figure 1 would be too complex. Also, the new figure about the use of OMVs in cancer therapy has been added.

  1. A careful review of spelling and grammar check should be carried out. There are numerous mistakes like “immunomodulation” spelling is wrong in Figure 1.

Reply: Spelling of  “immunomodulation”  was corrected. English editing was performed by the MDPI English service.

  1. There are examples of whole-cell bacterial minicells being engineered and used for glioblastoma treatment which are now in clinical trials. These should be included also in the Introduction section.

Reply: Thank you very much for your remark. We have included the use of whole-cell bacterial minicells in glioblastoma treatment in the section 3

Reviewer 2 Report

The authors of the paper titled “Fighting cancer with bacteria and their toxins” have presented a review featuring the functions of bacteria, their toxins and their released extracellular vesicles in tackling cancer from therapeutic as well as immunomodulatory perspectives. The review is interesting to a broader audience and is fit for publication with minor modifications.

  1. The review has a lot of relevant tables but is lacking any pictorial depictions of the different mechanisms leading to cancer cell death. e.g- increase in intracellular ROS production by bacterial toxins, modulation of tumor microenvironment by different cytokines etc. This should be included.
  2. A careful review of spelling and grammar check should be carried out. There are numerous mistakes like “immunomodulation” spelling is wrong in Figure 1.
  3. There are examples of whole-cell bacterial minicells being engineered and used for glioblastoma treatment which are now in clinical trials. These should be included also in the Introduction section.

Author Response

Reviewer: 2

  1. The authors of the paper titled “Fighting cancer with bacteria and their toxins” have presented a review featuring the functions of bacteria, their toxins and their released extracellular vesicles in tackling cancer from therapeutic as well as immunomodulatory perspectives. The review is interesting to a broader audience and is fit for publication with minor modifications.

Reply: We would like to thank you very much for your helpful remarks and suggestions. We tried to consider all of them and introduce suitable changes in the revised manuscript. We hope that now the new version of the manuscript will be recommended for publication. The detailed answers are below.

  1. The review has a lot of relevant tables but is lacking any pictorial depictions of the different mechanisms leading to cancer cell death. e.g- increase in intracellular ROS production by bacterial toxins, modulation of tumor microenvironment by different cytokines etc. This should be included.

Reply: Different mechanisms by which the bacterial toxins act are added to Figure 1. The different toxins have different modes of action to destroy the cancer cells, and therefore we have chosen to write them rather than show them pictorially, as Figure 1 would be too complex. Also, the new figure about the use of OMVs in cancer therapy has been added.

  1. A careful review of spelling and grammar check should be carried out. There are numerous mistakes like “immunomodulation” spelling is wrong in Figure 1.

Reply: Spelling of  “immunomodulation”  was corrected. English editing was performed by the MDPI English service.

  1. There are examples of whole-cell bacterial minicells being engineered and used for glioblastoma treatment which are now in clinical trials. These should be included also in the Introduction section.

Reply: Thank you very much for your remark. We have included the use of whole-cell bacterial minicells in glioblastoma treatment in the section 3.